# Epidemiology and Outcomes of Bloodstream Infections in HIV-Patients during a 13-Year Period

**DOI:** 10.3390/microorganisms8081210

**Published:** 2020-08-08

**Authors:** E. Franceschini, Antonella Santoro, Marianna Menozzi, Erica Bacca, Claudia Venturelli, Stefano Zona, Andrea Bedini, Margherita Digaetano, Cinzia Puzzolante, Marianna Meschiari, Gianluca Cuomo, Gabriella Orlando, Mario Sarti, Giovanni Guaraldi, Alessandro Cozzi-Lepri, Cristina Mussini

**Affiliations:** 1Infectious Disease Clinic, Azienda Ospedaliero-Universitaria di Modena, 41125 Modena, Italy; antonella.santoro7@gmail.com (A.S.); marymenozzi@gmail.com (M.M.); erica.bacca@gmail.com (E.B.); andreabedini@yahoo.com (A.B.); margheritadigaetano88@gmail.com (M.D.); cinzia.puzzolante@gmail.com (C.P.); mariannameschiari1209@gmail.com (M.M.); gian.cuomo@gmail.com (G.C.); gabriella.orlando7@virgilio.it (G.O.); giovanni.guaraldi@unimore.it (G.G.); crimuss@unimore.it (C.M.); 2Unit of Microbiology and Virology, Azienda Ospedaliero-Universitaria di Modena, 41125 Modena, Italy; venturelli.claudia@aou.mo.it (C.V.); sarti.mario@aou.mo.it (M.S.); 3Primary Care Department, AUSL Modena, 41125 Modena, Italy; ste.zona@gmail.com; 4Research Department of Infection & Population Health, Royal Free and University College Medical School, London 41125, UK; a.cozzi-lepri@ucl.ac.uk

**Keywords:** bloodstream infections, HIV, multidrug resistant

## Abstract

No data on antibiotic resistance in bloodstream infection (BSI) in people living with HIV (PLWH) exist. The objective of this study was to describe BSI epidemiology in PLWH focusing on multidrug resistant (MDR) organisms. A retrospective, single-center, observational study was conducted including all positive blood isolates in PLWH from 2004 to 2017. Univariable and multivariable GEE models using binomial distribution family were created to evaluate the association between MDR and mortality risk. In total, 263 episodes (299 isolates) from 164 patients were analyzed; 126 (48%) BSI were community-acquired, 137 (52%) hospital-acquired. At diagnosis, 34.7% of the patients had virological failure, median CD4 count was 207/μL. Thirty- and 90-day mortality rates were 24.2% and 32.4%, respectively. Thirty- and 90-day mortality rates for MDR isolates were 33.3% and 46.9%, respectively (*p* < 0.05). Enterobacteriaceae were the most prevalent microorganisms (29.8%), followed by Coagulase-negative staphylococci (21.4%), and *S. aureus* (12.7%). In BSI due to MDR organisms, carbapenem-resistant *K. pneumoniae* and methicillin-resistant *S. aureus* were associated with mortality after adjustment for age, although this correlation was not confirmed after further adjustment for CD4 < 200/μL. In conclusion, BSI in PLWH is still a major problem in the combination antiretroviral treatment era and it is related to a poor viro-immunological status, posing the question of whether it should be considered as an AIDS-defining event.

## 1. Introduction

HIV infection has various effects on cellular and humoral immunity, such as defective cell-mediated immunity and altered B-cell function. Furthermore, qualitative and quantitative neutrophil deficits and skin and mucous membrane defects predispose persons living with HIV (PLWH) to develop bacterial infections more frequently than a HIV-seronegative patient [1]. Infections in this population could be severe, leading not only to prolonged hospitalization and mortality but also to admittance in an intensive care unit (ICU) [2,3].

Indeed, in the post combination antiretroviral treatment (cART) era, PLWH are more frequently admitted to ICU due to blood stream infections (BSIs) than to *Pneumocystis jiroveci* pneumonia [4].

According to recent literature [5], clinical manifestations of BSIs in PLWH are similar to those in HIV seronegative patients; however, BSI incidence and mortality are often higher in HIV population [6]. Nowadays, bacterial infections account for 15% of mortality causes among HIV patients in the United States [7].

Concerning BSI epidemiology, differences have been described according to different geographical distribution of pathogens and access to cART. Indeed, in industrialized countries, cART has determined a reduction in incidence and a change of BSI clinical characteristics. In particular, reported predominant pathogens are Non-typhoidal *Salmonella*, *S. aureus*, *S. pneumoniae* and Coagulase-negative staphylococci (CoNS), with few data on antibiotic resistance profile [8].

Risk factors for BSIs in PLWH are high HIV-RNA, low CD4 cell count and a concomitant AIDS-defining condition [4]. Unfortunately, although current guidelines recommend the ‘test and treat’ strategy [9], still late presentation represents a major clinical problem in HIV infection. Indeed, some BSI etiologies such as Non-typhoidal *Salmonella*, Mycobacteria, and Cryptococci represent AIDS-defining conditions and their incidence is strictly linked to late presentation.

Finally, many PLWH become older in the post cART era, with an increase in comorbidities typically seen in the elderly, such as cardiovascular disease, hypertension, diabetes, chronic kidney disease, and non-AIDS defining cancers.

In this scenario, we expect to observe an increased risk of hospitalization and exposure to nosocomial infections in ageing HIV-infected populations, similar to that observed in HIV-negative patients.

Regarding nosocomial infections, in recent years the prevalence of multidrug-resistant (MDR) bacteria isolated from clinical samples continues to increase globally and Italy is one of the European countries with the highest incidence of MDR [10,11,12].

The objective of this analysis was to describe BSI epidemiology in PLWH in a single center but across a long period of time in the post cART era, focusing on antibiotic resistance data.

## 2. Materials and Methods

We performed a retrospective single-center observational study including all positive blood isolates in HIV-infected patients older than 18 years and admitted to Azienda Ospedaliero-Universitaria of Modena, Italy, from 1 January 2004 to 31 December 2017. Azienda Ospedaliero-Universitaria of Modena is a 1100-bed tertiary teaching hospital, with an average of 46,000 admissions per year.

The study was approved by the local ethical committee (protocol number 15592, 9th, March 2016). Informed consent was avoided according to Privacy Guaranteeing injunction nr. 85, 1st, March 2012, art. 4.2.

Patient and microbiology data were collected from computerized a clinical database.

BSI diagnosis criteria were as follows: blood cultures positive for pathogen bacteria; CoNS, *Corynebacterium* spp. and *Propionibacterium* spp. were regarded as contaminants, unless isolated from two or more separate blood culture sets.

Repeated positive blood cultures for the same organism in the 14 days after the first episode were excluded.

Hospital-acquired BSI was defined as a positive blood culture obtained on day 3 or later from hospital admission [13].

Polymicrobial BSI was defined as isolation of more than one bacterial species from the same blood cultures.

Bacterial resistance tests were interpreted in accordance with CLSI criteria until 2009 and with EUCAST break points from 2010. MDR, extensively drug-resistant (XDR) and pandrug-resistant bacteria were defined according to ECDC consensus 2012 [14].

Virological failure was defined as an HIV-RNA > 200 copies/mL.

We evaluated the following patient characteristics: age, gender, CD4, CD8, CD4/CD8 ratio, HIV-RNA, and cART.

Mean (SD) or median (IQR) were reported for continuous normally or non-normally distributed variables and compared between groups using appropriate parametric and non-parametric tests; categorical variables were described using frequencies and percentages and compared between groups using the chi-square test.

Univariable and multivariable Generalised Estimating Equation (GEE) models using binomial distribution family were fitted to evaluate the association between detection of MDR and the mortality risk. Patients with *Candida* BSI were excluded from the GEE analysis because Gram classification was not possible leading to missing data for this variable.

Statistical significance was defined on the basis of a *p*-value < 0.05. All statistical analyses were performed by using STATA 13.1 for Windows (StataCorp, College Station, TX, USA).

## 3. Results

A total of 263 episodes, corresponding to 299 isolates and obtained from 164 patients, from 1 January 2004 to 31 December 2017 were retrospectively analyzed. Ninety (34%) patients had more than one episode.

Community-acquired BSI represented 126 (48%) cases, while 137 (52%) were hospital-acquired. Twenty-four (9%) were polymicrobial. Of the 263 episodes, 218 (82.8%) were collected in the internal medicine wards, 22 (8.4%) in ICU, and 23 (8.8%) in surgical wards; 72 (27.4%) BSI originated from central indwelling catheters (CVC) (4/9 in ICU, 40%).

Patient characteristics at BSI diagnosis, stratified by detection of antibiotic resistance, are shown in Table 1.

Concerning HIV immune-virologic status, data were available for 172 episodes only. The majority of patients were on effective cART (65.3%). Nevertheless, 34.7% of patients with BSI had a virological failure, in particular 25% and 13% of patients presented HIV-RNA above 10,000 copies/mL and 100,000 copies/mL, respectively. Moreover, median CD4 count at the moment of the episode was low (207 CD4/μL).

According to patient history and clinical files, patients presented a previous cART initiation in 205 episodes (88.4%, data available for a total of 232 episodes). Twenty-one patients, for a total of 27 episodes (11.6%), had never been prescribed antiretroviral treatment before BSI.

A total of 83 patients (48%) died, with a median survival time of 28 (IQR 6-274) days after the last episode of bacteremia. It is noteworthy that 16 (19.3%) out of 83 patients died more than one year after the last episode. Thirty-day mortality rate after the last episode was 24.2% while 90-day mortality rate was 32.4%.

In particular, mortality rates for MDR isolates were 33.3% (22 patients) and 46.9% (31 patients) at 30 and 90-days after last episode of bacteremia (both *p*-value < 0.05), respectively. No statistical difference was found in median time to death after the last episode of bacteremia according to the presence of specific MDR isolates.

Crude mortality rate for BSI was significantly higher for MDR isolates as compared to non-MDR isolates (44.7% versus 24.3%, respectively).

Figure 1 and Figure 2 show the distribution of BSI isolates in the entire hospital and in the different hospital wards, respectively.

In the entire PLWH population, Enterobacteriaceae were the prevalent microorganisms (88, 29.8%), followed by ConS (63, 21.4%), and *S. aureus* (38, 12.7%); non-fermenting Gram-negative bacteria were 37 (12.4%), Streptococci were 24 (8%) and Enterococci 22 (7.4%). Among Streptococci, *S. pneumoniae* was the most frequent pathogen (11, 45.8%), followed by *S. viridans* group (5, 20.8%). Among Enterobacteriaceae, *E. coli* was the most frequent pathogen (49, 55.7%), followed by *K. pneumoniae* (19, 21.6%) and *E. cloacae* (8, 9.1%).

Among non-fermenting Gram-negative organisms, the majority was *P. aeruginosa* (21, 56.8%), followed by *A. baumannii* (4, 10.8%). *E. faecalis* was the most frequent organism among Enterococci (11, 50%); *E. faecium* were 10 (45.5%). Among Staphylococcal bacteremia, CoNS were the most frequently isolated (63, 62.4%) *S. aureus* was isolated in 38 cases (37.6%). CoNS accounted for 36% of CVC-related BSIs (vs. peripheral lines *p* < 0.001). Staphylococcal isolates were resistant to methicillin in 63.4% of cases. *Candida* spp. were 7 (2.3%).

Among community-acquired BSIs, Gram negative were the most frequent pathogens, while among nosocomial-acquired BSIs, Gram positive were the most prevalent (*p* = 0.001). No candidemia was observed among community-onset BSIs. Regarding the different wards, Gram-positive isolates were more frequent in the ICU and in medical wards, while Gram-negative isolates were prevalent in surgical wards (*p* = 0.031). This higher prevalence of Gram-positive isolates in ICU could be related to the enrichment of BSI from central indwelling lines.

Fifty-nine patients (34.1%) presented at least one bacteremia due to a MDR/XDR bacterium, accounting for a total of 96 BSIs. In particular, 11 out of 38 (28.9%) *S. aureus* were methicillin-resistant (MRSA). Extended spectrum beta-lactamases (ESBL) were present in 21 out of 88 Enterobacteriaceae (23.9%); in particular, in 7 out of 49 (14.3%) *E. coli* and 9 out of 19 (47%) *K. pneumoniae*. Five out of 19 (26.3%) *K. pneumoniae* were carbapenemase-resistant (CRKP) and 6 out of 29 (20.7%) *P. aeruginosa* were resistant to carbapenems, while 1 out of 22 Enterococci was resistant to vancomycin (4.6%).

Figure 3 shows the proportion of MDR/XDR isolates by calendar period of observation.

Concerning the resistance pattern, there was no evidence for a difference in proportion of MDR isolates by calendar period of observation. Only the proportion of ESBL Enterobacteriaceae increased significantly over the period 2014–2017 (*p* 0.038).

Table 2 shows the estimates from fitting an univariable GEE logistic regression model for all factors considered in the analysis in the entire population, in Gram-negative BSIs and in Gram-positive BSIs, respectively. In this unadjusted analysis, age, years of cART, BSI due to MDR organisms, and BSI due to CRKP were all significantly associated with mortality risk in the entire population. The same factors were confirmed when restricting the analysis to Gram-negative BSI. In contrast, in the subpopulation of Gram-positive BSI, only years of cART was significantly associated with the risk of death. However, these are subset analyses so *p*-values need to be interpreted with caution.

Table 3A–C show GEE models (unadjusted and adjusted) for factors associated with mortality in BSI due to MDR, CRKP and MRSA isolates, respectively.

Of note, BSI due to MDR organisms was associated with a twofold higher risk of death after adjustment for age even if this correlation was not confirmed after further adjustment for CD4 < 200/μL. As shown in the table, this result was confirmed in BSI due to CRKP and in BSI due to MRSA (although results were more compatible with the null hypothesis of no difference).

HIV-associated variables did not associate with the mortality outcome in any of the analyses.

## 4. Discussion

To our knowledge this is the first study that describes BSI epidemiology in PLWH in the setting of high MDR prevalence. Obviously, epidemiology and antibiotic resistance profile can vary widely among different hospitals so it is very important to describe patterns of BSI in patients seen for care at our institution in order to prescribe empirical antibiotic treatment.

In our study, Enterobacteriaceae (in particular *E. coli*) were the most frequent organisms involved in BSI (29.8%) and this is in line with data obtained from the general population of the same region (RER). Several studies showed a higher prevalence of Gram-positive bacteria in nosocomial BSI in PLWH, with a proportion of Gram negative ranging from 20% to 31% [6,8,15]. Indeed, among patients with hospital-onset BSI and patients admitted to the ICU, Gram-positive bacteria, particularly CoNS, are the most prevalent pathogens. This finding could be explained by the higher prevalence of BSI from central indwelling lines in this setting.

Concerning the resistance pattern, the proportion of detected MDR isolates did not differ significantly over the years. Only the proportion of ESBL Enterobacteriaceae increased significantly in the period 2014–2017. Comparing our resistance pattern to that in the RER [16] we can observe a similar proportion of MRSA, a lower proportion of CRKP, and a slightly higher proportion of ESBL. Actually, our data show that the BSI epidemiology in PLWH does not differ much from that of the general population.

Regarding MRSA bacteremia, in our population of PLWH the percentage was 28.9%. This prevalence is lower than expected as MRSA is usually considered more frequent in PLWH than in the general population. Indeed, a recent study of Furuno et al. observed 66% of MRSA among *S. aureus* bacteremia and studies suggested PLWH to be at increased risk of community-acquired MRSA because of overlapping community networks as well as the high prevalence of intravenous drug use [17].

Concerning HIV-related viro-immunological conditions, PLWH included in our study had a poor status with a median of 207 CD4 cells/μL and a detectable HIV-viral load (VL) in 34.7% indicating not only late presentation, but also, among those on cART treatment, either virological failure or low adherence. Indeed, the study population has a worse immune-virological situation compared to the average PLWH followed in the cART era in our clinic as typically more than 90% of patients on cART have an undetectable HIV-VL [18]. This finding, although cross-sectional, is very relevant and suggests that PLWH with poor adherence or a recent AIDS diagnosis are at increased risk of BSI. Studies evaluating the role of CD4 count as risk factor for BSI in PLWH showed conflicting results. On one hand, two studies described low CD4 counts in all hospitalized HIV patients, but they found no significant difference in patients with and without BSI [15,19]. On the other hand, other three studies described a significant increased incidence of bacterial BSI in HIV-infected patients with lower CD4 counts compared to those with a higher CD4 counts [20,21,22].

Despite the increased risk of BSI, HIV viro-immunological status was not associated with mortality risk in our analysis, suggesting that they may play less of a role in the setting of people with BSI. Indeed, CRKP bacteremia was a risk factor for mortality, while MRSA showed only a trend for an association. In line with our findings, Furuno et al. showed a major role of MRSA bacteremia compared to HIV viro-immunological status on mortality.

In our study the 30-and the 90-day mortality rates were similar to what has been described in both PLWH and HIV-negative patients [6,23,24,25,26]. Mortality for BSI due to MDR isolates was significantly higher than that observed in people without MDR (44.7% versus 24.3%, respectively).

Our study has some limitations: first of all, it is a retrospective study, thus unmeasured confounding factors cannot be ruled out. Indeed, some key potential confounders such as viro-immunological characteristics, the prevalence of intravenous drug users, and antibiotic treatment were missing. A correct inference also relies on a correct specification of the model and inclusion of all the important measured confounding factors.

Nevertheless, we think that this study has some strengths: it is the first study conducted in an MDR organisms endemic country and the observation period was very long period of time (13 years).

In conclusion, BSI in PLWH is still a major problem in the cART era and appears to be related to a poor viro-immunological status. This underlines the importance of an early HIV diagnosis and a good retention in care in order to strengthen a good adherence to cART. Importantly, now National and International Guidelines [27,28] recommend in PLWH vaccinations against invasive bacterial diseases, such as S. *pneumoniae* or *H. influenzae* so BSI incidence might decrease in the future as a consequence of this intervention.

Finally, the implementation of antimicrobial stewardship programs and infection control strategies remains key to reduce MDR BSI and protect frail populations from MDR infections.

## Figures and Tables

**Figure 1 microorganisms-08-01210-f001:**
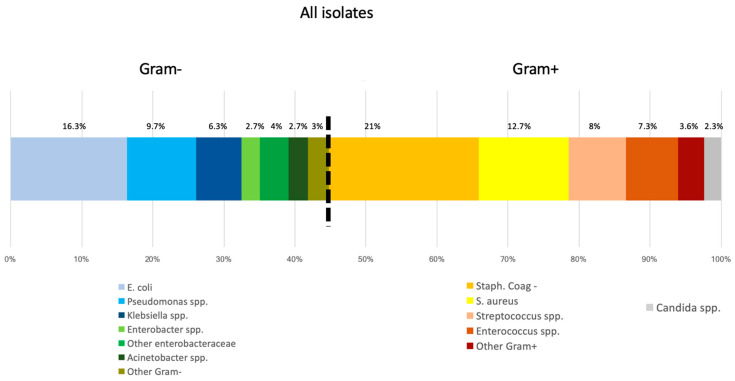
Microbiological distribution of bloodstream infection (BSI) isolates in the whole hospital according to Gram stain and origin of BSI (nosocomial or hospital acquired).

**Figure 2 microorganisms-08-01210-f002:**
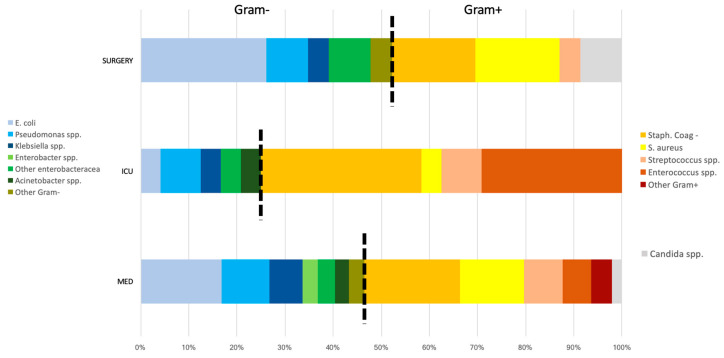
Distribution of BSI isolates in the different hospital wards according to Gram stain.

**Figure 3 microorganisms-08-01210-f003:**
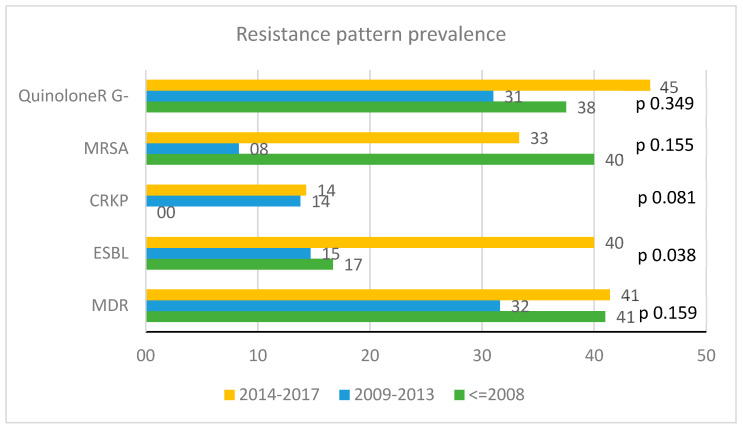
Proportion of multidrug-resistant (MDR)/extensively drug resistant (XDR) isolates by calendar period. Rates of resistance of Quinolone R, carbapenem-resistant *K. pneumoniae* (CRKP) and extended-spectrum beta-lactamase (ESBL) pattern are defined within Enterobacteriaceae. Rates of methicillin-resistant *S. aureus* (MRSA) are defined within Staphylococcal infections. MDR: multidrug resistant; ESBL: extended-spectrum beta-lactamase; CRKP: carbapenem-resistant *K. pneumoniae*; MRSA: methicillin-resistant *S. aureus*.

**Table 1 microorganisms-08-01210-t001:** Patient characteristics at bloodstream infection diagnosis.

Patient Characteristics	Total (263 Obs)	Non MDR Isolate (169 Obs)	MDR Isolate (94 Obs)	*p* Value
Age, median (IQR)	46 (40–52)	46 (41–52)	45 (40–52)	0.625
Male sex, *n* (%)	175 (66.5)	114 (67)	61 (65)	0.673
Year of event, *n* (%)				0.188
Before 2008	105 (40)	62 (36.7)	43 (45.7)	
2009–2013	94 (35.7)	67 (39.6)	27 (28.7)	
After 2014	64 (24.3)	40 (23.7)	24 (25.5)	
Hospital acquired, *n* (%)	137 (52)	71 (42)	66 (70)	<0.001
Years of HIV *, median (IQR)	14 (7–22)	15 (8–23)	11 (4–19)	0.021
Years of cART **, median (IQR)	5 (0–10)	5 (0–10)	5 (0–9)	0.944
CD4/CD8 ratio ***, median (IQR)	0.38 (0.13–0.64)	0.41 (0.14–0.66)	0.24 (0.10–0.60)	0.133
CD4 count ***, cell/μL, median (IQR)	207 (73–385)	216 (84–396)	186 (45–344)	0.356
CD4 count < 200/μL at event ***, *n* (%)	83 (47.4)	52 (45)	31 (52)	0.417
HIV RNA > 100.000 copies/mL at event ***, *n* (%)	23 (13.3)	12 (10.8)	11 (17.7)	0.198
HIV RNA > 10.000 copies/mL at event ***, *n* (%)	44 (25.4)	28 (25.2)	16 (25.8)	0.933
HIV RNA < 200 copies/mL at event ***, *n* (%)	113 (65.3)	75 (67)	38 (61)	0.406
Type of isolate, *n* (%)				0.061
Gram negative	119 (45.2)	82 (48.5)	37 (49.4)	
Gram positive	139 (52.8)	82 (48.5)	57 (60.6)	
Fungi	5 (2)	5 (3)	0 (0)	
Death, *n* (%)	83 (31.6)	41 (24.3)	42 (44.7)	<0.001
Gram negative	39 (32.8)	20 (24.4)	19 (51.4)	0.004
Gram positive	43 (30.9)	20 (24.4)	23 (40.4)	0.045
Fungi	1 (20)	1 (20)	0 (0)	-
Polymicrobial, *n* (%)	24 (9.1)	15 (8.9)	9 (9.6)	0.854
Central venous associated bacteremia, *n* (%)	72 (27.4)	34 (20.1)	38 (40.4)	<0.001
Gram negative	26 (36.1)	14 (41.2)	12 (31.6)	0.183
Gram positive	44 (61.1)	18 (52.9)	26 (68.4)	
Fungi	2 (2.8)	2 (5.9)	0 (0)	
ICU	9 (40.9)	5 (14.7)	4 (10.5)	0.849
Surgery	6 (26.1)	3 (8.8)	3 (7.9)	
Medicine	57 (26.2)	26 (76.5)	31 (81.6)	

*n*: number; IQR: interquartile range; cART: combination antiretroviral treatment; HIV: human immunodeficiency virus; VL: viral load; MDR: multidrug resistant; ESBL: extended-spectrum beta-lactamase; CRKP: carbapenem-resistant *K. pneumoniae*; MRSA: methicillin-resistant *S. aureus*. * data available for 244 episodes only; ** data available for 229 episodes only; *** data available for 173 episodes only.

**Table 2 microorganisms-08-01210-t002:** Univariable GEE model for factors associated with mortality.

Variables	Total Population (258 Obs)				Gram Negative (119 Obs)				Gram Positive (139 Obs)			
	OR	Std. Err.	*p* Value	95% CI	OR	Std. Err.	*p* Value	95% CI	IRR	Std. Err.	*p* Value	95% CI
Age	1.04	0.02	0.009	1.01–1.07	1.08	0.03	0.004	1.02–1.14	1.02	0.01	0.249	0.99–1.06
Male sex	1.15	0.36	0.639	0.63–2.11	0.56	0.24	0.187	0.24–1.32	2.29	1.03	0.066	0.95–5.55
Years of c-ART	1.12	0.03	0.001	1.06–1.19	1.10	0.04	0.006	1.03–1.18	1.13	0.05	0.004	1.04–1.24
Years of HIV	1.02	0.16	0.331	0.98–1.05	1.03	0.02	0.176	0.99–1.08	1.00	0.02	0.872	0.95–1.04
CD4 count < 200 at event	0.67	0.23	0.255	0.34–1.33	1.09	0.55	0.859	0.41–2.94	0.39	0.19	0.050	0.16–1.00
HIV-RNA < 200 copies/mL at event	0.96	0.34	0.912	0.48–1.85	1.04	0.54	0.932	0.38–2.89	0.91	0.43	0.844	0.36–2.29
HIV-RNA > 100.000 copies/mL at event	0.64	0.35	0.410	0.22–1.83	0.43	0.36	0.317	0.08–2.25	0.73	0.52	0.657	0.18–2.93
HIV-RNA > 10.000 copies/mL at event	0.78	0.31	0.538	0.36–1.70	0.50	0.32	0.278	0.14–1.74	1.00	0.50	0.997	0.37–2.68
Hospital ward												
ICU	Ref	Ref	Ref	Ref	Ref	Ref	Ref	Ref	Ref	Ref	Ref	Ref
Surgery	0.32	0.20	0.075	0.09–1.12	1.04	0.13	0.072	0.01–1.22	0.16	0.19	0.118	0.02–1.58
Medicines	0.44	0.20	0.078	0.18–1.09	1.09	0.12	0.048	0.01–1.00	0.56	0.31	0.295	0.19–1.65
CVC-associated bacteremia	1.16	0.32	0.582	0.68–2.00	0.94	0.41	0.892	0.40–2.22	1.28	0.49	0.521	0.60–2.72
MDR isolate	2.46	0.65	0.001	1.46–4.13	3.37	1.41	0.004	1.48–7.65	2.12	0.78	0.041	1.03–4.34
ESBL isolate	1.83	0.63	0.077	0.94–3.59	1.57	0.61	0.249	0.73–3.37	-	-	-	-
CRKP isolate	5.22	2.99	0.004	1.70–16.08	4.02	2.18	0.010	1.39–11.65	-	-	-	-
MRSA isolate	3.48	2.22	0.051	1.00–12.15	-	-	-	-	3.66	2.18	0.061	0.94–12.01

Obs: observations; OR: odd ratio; CI: confidence interval; c-ART: combination antiretroviral treatment; HIV: human immunodeficiency virus; VL: viral load; ICU: intensive care unit; CVC: central venous catheter; MDR: multidrug resistant; ESBL: extended-spectrum beta-lactamase; CRKP: carbapenem-resistant *K. pneumoniae*; MRSA: methicillin-resistant *S. aureus;* GEE: Generalised Estimating Equation

**Table 3 microorganisms-08-01210-t003:** A: Multivariable GEE models for factors associated with mortality in the entire population; B: CRKP isolate subanalysis; C: MRSA isolate subanalysis.

**A**	**Unadjusted (258 Obs)**			**Adjusted for Age (258 Obs)**			**Adjusted for Age, Ward and CD4 Count * (172 Obs)**		
	**OR**	**95%CI**	***p* Value**	**OR**	**95%CI**	***p* Value**	**OR**	**95%CI**	***p* Value**
MDR isolate	2.49	1.49–4.17	0.001	2.55	1.50–4.34	0.001	2.22	1.12–4.41	0.023
Age				1.04	1.01–1.07	0.009	1.04	1.00–1.07	0.046
**B**	**Unadjusted (258 Obs)**			**Adjusted for Age (258 Obs)**			**Adjusted for Age, Ward and CD4 Count * (172 Obs)**		
	**OR**	**95%CI**	***p* Value**	**OR**	**95%CI**	***p* Value**	**OR**	**95%CI**	***p* Value**
CRKP isolate	5.22	1.70–16.08	0.004	4.45	1.39–14.22	0.012	1.84	0.46–7.39	0.388
Age				1.03	1.00–1.06	0.035	1.03	1.00–1.07	0.074
**C**	**Unadjusted (258 Obs)**			**Adjusted for Age (258 Obs)**			**Adjusted for Age, Ward and CD4 Count * (172 Obs)**		
	**OR**	**95%CI**	***p* Value**	**OR**	**95%CI**	***p* Value**	**OR**	**95%CI**	***p* Value**
CRKP isolate	5.22	1.70–16.08	0.004	4.45	1.39–14.22	0.012	1.84	0.46–7.39	0.388
Age				1.03	1.00–1.06	0.035	1.03	1.00–1.07	0.074

(**A**) * Model adjusted for hospital ward and CD4 count < 200 cells/mcl at event (*p*-value > 0.05). Obs: observations; OR: odd ratio; CI: confidence interval; MDR: multidrug resistant; (**B**) * model adjusted for hospital ward and CD4 count < 200 cells/mcl at event (*p*-value > 0.05). Obs: observations; OR: odd ratio; CI: confidence interval; MDR: multidrug resistant; CRKP: carbapenem-resistant *K.pneumoniae*; (**C**) * model adjusted for hospital ward and CD4 count < 200 cells/mcl at event (*p*-value > 0.05). Obs: observations; OR: odd ratio; CI: confidence interval; MRSA: methicillin-resistant *S. aureus*; GEE: Generalised Estimating Equation.

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
