# Peer review of "Epidemiology and Outcomes of Bloodstream Infections in HIV-Patients during a 13-Year Period"

_microorganisms, 2020, doi:10.3390/microorganisms8081210_

Round 1
Reviewer 1 Report
This is an interesting study on the types of bloodstream infections that occur in patients living with HIV. It is significant to the study of HIV and infectious disease.
A minor comment is on the presentation style. The manuscript is very dry and as such, is a bit difficult to read. There are far too many acronyms (especially in the abstract), some of which are not spelled out. This makes it difficult to follow the text. The results section, while very clear, is dry and offers no insight into the relevance. It is recommended that some of the less subjective points from the discussion be moved to the results section to increase interest.
Author Response
The manuscript is very dry and as such, is a bit difficult to read. There are far too many acronyms (especially in the abstract), some of which are not spelled out. This makes it difficult to follow the text. The results section, while very clear, is dry and offers no insight into the relevance. It is recommended that some of the less subjective points from the discussion be moved to the results section to increase interest.
We revised all the manuscript trying to obtain a more fluent english. We removed different acronyms and we spelled out the missing ones.
We tried to underline the relevant points in the results.
Please see the attached document.

Reviewer 2 Report
It is very interesting artickle explaining the role of pathogens and their resistance, closely connected with bloodstreem infections in patients with HIV/AIDS.
We were missing these informations already for long time.
Author Response
thank you for your comments
